# Anti-*Heliobacter pylori* and Anti-Inflammatory Potential of *Salvia officinalis* Metabolites: In Vitro and In Silico Studies

**DOI:** 10.3390/metabo13010136

**Published:** 2023-01-16

**Authors:** Hatun A. Alomar, Wafaa M. Elkady, Marwa M. Abdel-Aziz, Taghreed A. Ibrahim, Noha Fathallah

**Affiliations:** 1Pharmacology and Toxicology Department, College of Pharmacy, King Saud University, Riyadh 11451, Saudi Arabia; 2Department of Pharmacognosy and Medicinal Plants, Faculty of Pharmacy, Future University in Egypt, Cairo 11835, Egypt; 3Regional Center for Mycology and Biotechnology (RCMB), Al-Azhar University, Cairo 11651, Egypt; 4Department of Pharmacognosy, College of Pharmacy, King Saud University, Riyadh 11451, Saudi Arabia; 5Department of Pharmacognosy, Faculty of Pharmacy, Cairo University, Cairo 11562, Egypt

**Keywords:** *Salvia officinalis*, *Helicobacter pylori*, ethanolic extract, essential oil, antibacterial, anti-inflammatory, Glucose-6 phosphatase

## Abstract

Due to its rising antibiotic resistance and associated inflammations, *Helicobacter pylori* poses a challenge in modern medicine. *Salvia officinalis*, a member of the Lamiaceae family, is a promising medicinal herb. In this regard, a phytochemical screening followed by GC-MS and LC-MS was done to evaluate the chemical profile of the total ethanolic extract (TES) and the essential oil, respectively. The anti-*H. pylori* and the anti-inflammatory activities were evaluated by a micro-well dilution technique and COX-2 inhibition assay. Potential anti-*H. pylori* inhibitors were determined by an in silico study. The results revealed that the main metabolites were flavonoids, sterols, volatile oil, saponins, and carbohydrates. The LC-MS negative ionization mode demonstrated 12 compounds, while GC-MS showed 21 compounds. Carnosic acid (37.66%), epirosmanol (20.65%), carnosol1 (3.3%), and 12-*O*-methyl carnosol (6.15%) were predominated, while eucalyptol (50.04%) and camphor (17.75%) were dominant in LC-MS and GC-MS, respectively. TES exhibited the strongest anti-*H. pylori* activity (3.9 µg/mL) asymptotic to clarithromycin (0.43 µg/mL), followed by the oil (15.63 µg/mL). Carnosic acid has the best-fitting energy to inhibit *H. pylori* (−46.6769 Kcal/mol). TES showed the highest reduction in Cox-2 expression approaching celecoxib with IC_50_ = 1.7 ± 0.27 µg/mL, followed by the oil with IC_50_ = 5.3 ± 0.62 µg/mL. Our findings suggest that *S. officinalis* metabolites with anti-inflammatory capabilities could be useful in *H. pylori* management. Further in vivo studies are required to evaluate and assess its promising activity.

## 1. Introduction

Since the beginning of time, people have embraced remedies from plants, animals, or marine organisms to combat and prevent disease. Fossil evidence demonstrates that humans have been using plants as remedies for at least 60,000 years [1]. Herbal medicine, often known as phytotherapy, is the practice of treating patients with herbal treatments [2].

In the last decade, resistance to conventional antibiotics has made the treatment of bacteria a tremendous challenge to all healthcare providers. There has been a renaissance in the investigation of plants as a potential source for new and robust antibiotics due to rising global demand for novel natural or synthetic forms of treatment [3]. *Helicobacter pylori* (*H. pylori*) is a Gram-negative bacteria that were identified as a group I carcinogen by the International Agency for Research and Cancer [4]. The World Health Organization (WHO) estimated in 1994 that around half of the world’s population was infected with this micro-organism. Even though most infections are asymptomatic [5], *H. pylori* are blamed for over 550,000 new cases of stomach cancer each year. In Egypt and other developing countries, its prevalence could reach up to 90%, while in developed countries, the prevalence is below 40% [6]. The prevalence also varies according to age, socioeconomic and educational levels, occupation, and lifestyle [7,8]. There was usually an association between pathological outcomes such as peptic ulcers, intestinal metaplasia, chronic gastritis, gastric lymphoma, or cancer development in chronically infected patients [8]. Soon, *H. pylori* infection is expected to be among the top 10 global causes of death. The popular medications metronidazole (MTZ), clarithromycin (CLA), amoxicillin, and tetracycline are used to treat *H. pylori* infection. However, the mutations of the bacteria can result in antibiotic resistance, particularly to CLA and MTZ. Second-line treatment strategies need to be developed as soon as possible to overcome this resistance [9]. One of the most fruitful strategies suggested was the combination of different antibiotics with other drugs that can be natural or synthetic [10].

The Labiatae family, usually referred to as the mint family, contains a vital group of medicinal plants. The members of this family are usually herbs or shrubs with strong aromatic odors. They are widespread in the Mediterranean regions [11]. The mint family includes numerous plants with reported antibacterial activities such as *Origanum vulgare* (oregano), *Thymus vulgaris* (thyme), *Rosmarinus officinalis* (rosemary), *Mentha piperita* (mint), and *Salvia officinalis* (sage). *Salvia* is the family’s largest genus, with around 900 species [12,13,14]. Sage is named “Salvation Plant” as the name comes from the Latin word “salvarem”, which means “to save or cure” [2]. Herbalist Sebastian Kneipp once said, “The one who has the garden, should plant sage because when you got that—it is natural pharmacy always at hand” [1]. *S. officinalis* is native to the Mediterranean region, although it has been naturalized in various locations around the globe, and many species are now grown in Egypt [15]. It has been used to reduce sweating, as a gargle for a sore throat, to improve menstrual cycle regularity, to decrease hot flashes in menopause, to fight gastroenteritis, to improve lipid status, to improve appetite and digestion, and to improve mental capacity [2,14]. Recently, it has been reported that essential oil can be used in the treatment of Alzheimer’s disease, and it has also the potential in treating cancer as it shows strong antitumorigenic activities [16]. Many studies were done on the antimicrobial activity of sage against different strains of bacteria such as *S. aureus* [17], periodontopathogens [18], and fungi [17,19,20,21]. Some metabolites of *S. officinalis* extracts and essential oils were proven to have strong antibacterial properties [22,23], and thus, it was recognized as a plant with significant potential as an *H. pylori* medication [24].

The primary phytochemicals metabolites illustrated before in *S. officinalis* aerial parts were flavonoids; carbohydrates; fatty acids; glycosidic derivatives (e.g., cardiac glycosides, flavonoid glycosides, and saponins); phenolic compounds (e.g., coumarins, flavonoids, and tannins); polyacetylenes; steroids; and terpenes/terpenoids (e.g., monoterpenoids and diterpenoids), as well as volatile constituents such as borneol, cineole, camphor, and thujone [14].

Our current research intends to screen the active metabolites present in sage’s aerial parts, extract the essential oil and the total ethanolic extract (TES), and analyze the chemical profiles of the oil and extract using GC-MS and LC-MS, respectively. To assess and compare the anti-*H. pylori* and anti-inflammatory effects of them both. Finally, to determine the main constituents with the highest potential as anti-*H. pylori* agents using in silico analysis.

This study is a continuation of the author’s investigation into numerous herbal remedies against this specific organism [3,25]. To our realization, this is the first study to compare the antibacterial effects of *S. officinalis* oil and extract against *H. pylori*.

## 2. Experimental Design

### 2.1. Plant Material

*Salvia officinalis* L. aerial parts (1000 g) were collected from farmlands of the Medicinal, Aromatic and Poisonous Plants Experimental Station of the Faculty of Pharmacy, Cairo University, Giza, Egypt. The aerial parts were identified and cleaned in water, air-dried, pulverized, and stored in well-sealed containers at room temperature until needed.

### 2.2. Phytochemical Screening

The dried aerial parts of *S. officinalis* (20 g) were subjected to preliminary phytochemical screening as described by [26,27] to identify various phytochemical components present in it; flavonoids, carbohydrates, tannins, saponins, sterols, and volatile oil. All the chemicals were acquired from Pio-chem firm in Cairo, Egypt, and were of excellent purity. Ferric chloride, HCL, Dragendorff’s reagent, methanol, chloroform, H_2_SO_4_, concentrated ammonia, ethanolic KOH, NaOH, and glacial acetic acid were among the compounds used as prescribed by [28,29].

### 2.3. Total Ethanolic Extract (TES)

The air-dried *S. officinalis* aerial parts (250 g) were extracted using 95% ethanol (3 × 500 mL) at room temperature until exhaustion, and the solvent was evaporated under reduced pressure at a temperature not exceeding 60 °C yielding (20 g) residue that was stored in amber-colored glass well-closed containers in a refrigerator until usage.

### 2.4. Essential Oil Extraction

According to the procedure outlined in the European Pharmacopoeia [30] and as described by [3,31], the dried *S. officinalis* aerial parts (250 g) were hydro-distilled for 5 h using a Clevenger apparatus (plant to water ratio: 1:3 *w/v*). The volume (mL) of essential oil for every 100 g of the investigated plant was used to compute the yield percentage, and the oily phases were maintained after being separated and dried over anhydrous sodium sulfate.

### 2.5. GC-MS Analysis for Essential Oil

Recording of the mass spectra was done using Shimadzu GCMS-QP2010 (Kyoto, Japan) attached to Rtx-5MS fused bonded column (30 m × 0.25 mm i.d. × 0.25 μm film thickness) (Restek, Las Vegas, NV, USA) with a split–split-less injector. The starting column temperature was kept at 45 °C for 2 min (isothermal) then programmed to 300 °C at a rate of 5 °C/min and kept constant at 300 °C for 5 min (isothermal). The injector temperature was 250 °C. The flow rate of the helium carrier gas was 1.41 mL/min. The recording of the mass spectra was done by applying the following condition: (system current) filament emission current, 60 mA; ionization voltage, 70 eV; and ion source, 200 °C. Diluted samples (1% *v*/*v*) were injected with split mode (split ratio 1:15).

### 2.6. LC/MS Analysis

The ethanolic extract was analyzed using UPLC/ESI-MS, ACQUITY UPLC System-Waters Corporation (Milford, MA, USA). The sample (100 µg/mL) was prepared using MeOH (HPLC grade) and filtered using a membrane disc filter (0.2 μm) before analysis. This was carried out in Column: ACQUITY UPLC-BEH C18 1.7 µm–2.1 × 50 mm. Injection volume: 10 μL. Solvent system: consisted of (A) Water containing 0.1% formic acid and (B) Methanol containing 0.1% formic acid. Elution was done using gradient mobile phase starting from (90% A: 10% B), (70% A: 30% B), (30% A: 70% B), (10% A: 90% B), and (0% A: 100% B) at a flow rate of 0.2 mL/min. The parameters for analysis were carried out using negative ion acquisition mode using XEVO TQD triple quadrupole, Waters Corporation, Milford, MA01757 USA, mass spectrometer. Source temperature 150 °C, cone voltage 30 eV, capillary voltage 3 kV, desolvation temperature 440 °C, cone gas flow 50 L/h, and desolvation gas flow 900 L/h. Mass spectra were detected in the ESI between m/z 100–1000. The peaks and spectra were processed using the Maslynx 4.1 software and tentatively identified by comparing their retention times (Rt) and mass spectra with reported data.

### 2.7. In Vitro Evaluation of Anti-H. pylori Activity

The antibacterial activity of the *S. officinalis* essential oil and the ethanolic extract was rapidly screened to confirm their previously recorded activity [32,33]. Colorimetric broth micro-dilution method using XTT [2,3-bis(2-methoxy-4-nitro-5-sulfo-phenyl)-2H-tetrazolium-5-carboxanilide] reduction assay was adopted to determine the MIC against the reference strain of *H. pylori* ATCC 43504 according to [3,34,35]. The MIC was specified as the extract concentration that produced a 100% decrease in optical density compared with control growth results. Clarithromycin was used as a standard antibacterial agent. The volatile oils were serially diluted in 5% (DMSO) Dimethyl sulfoxide solution containing 0.1% Tween 80 (*v*/*v*) (10 mg/mL), and then, 50 μL of each dilution was added to wells in a microtiter plate containing 100 μL TSB. Fifty microliters of adjusted microbial inoculum (10^6^ cells/mL) were added to each well, and then the microtiter plates were incubated in the dark at 37 °C for 24 h. After incubation, 100 μL of freshly prepared XTT were added and incubated again for 1 h at 37 °C [36]. Colorimetric variation in the XTT assay was measured using a microtiter plate reader at 492 nm.

### 2.8. In Silico Evaluation of Anti-H. pylori Activity

Molecular docking analysis (in silico) was done for the identified compounds in both the essential oil and the ethanolic extract regarding the active center of *H. pylori* glucose 6-phosphate dehydrogenase (HpG6PD), enzyme (PDB ID: 7SEH) was performed applying Discovery Studio 4.5 (Accelrys, Inc., San Diego, CA, USA). This enzyme is necessary for the metabolism of bacteria. The Protein Data Bank (http://www.pdb.org) accessed 5 December 2022, was used to obtain the protein (enzyme). The co-crystallized ligand nicotinamide-adenine-dinucleotide phosphate (NADP) was used to determine the docking binding site. The free binding energies were computed for the most stable docking locations of the co-crystalized ligand, as well as for all identified molecules.

### 2.9. Anti-Inflammatory Assay

To examine the anti-inflammatory response by inhibiting the cyclooxygenase-2 (COX-2) enzyme, samples with concentrations between 125.0 and 0.98 g/mL were analyzed.

The COX (EC 1.14.99.1) activity was assessed as the result of the oxidation reaction between *N*, *N*, *N*, *N*-tetramethyl-p-phenylenediamine (TMPD) and arachidonic acid. With a few minor modifications, this experiment was carried out following the previously reported procedure [37,38]. Using a microplate reader, the inhibitory action was assessed by observing the rise in absorbance at 611 nm (BIOTEK, Santa Clara, CA, USA). Inhibitory activity (%) = (1 As/Ac) x − 100, where (As) is the absorbance in the presence of the test drug and (Ac) is the absorbance of the control, was used to compute the inhibitory percentages. The concentration eliciting the most inhibition of the COX-2 isoenzyme was used to measure the effectiveness of the extracts and the reference drug (celecoxib).

## 3. Results

### 3.1. Phytochemical Screening

To assess a plant’s potential therapeutic value, as well as to identify the active ingredients responsible for the known biological activity displayed by plants, a phytochemical screening is crucial. Additionally, it offers the base for more accurate identification of compounds and more accurate research. The first step in this work was the screening of bioactive molecules. The screening revealed the presence of flavonoids, carbohydrates, tannins, saponins, sterols, and volatile oil as presented in Table 1.

### 3.2. LC/MS of the Ethanolic Extract

The LC/MS analysis illustrates the chemical profile of the total ethanolic extract (Figure 1 and Figure 2 and Table 2). The appropriate spectrum parameters, including molecular ions, mass spectra, and retention time, were used for identification.

### 3.3. Extraction and GC/MS of the S. officinal Essential Oil

Hydro distillation of *S. officinalis* aerial parts results in the production of (0.7%) essential oil yield, which exhibited a light-yellow color with distinct strong aromatic fragrances. The chemical profile of *S. officinalis* oil resulted in the identification of 21 compounds. Several compounds were previously reported but with different percentages that may depend on the season, the geographic origin, extraction methods, and environmental factors [49,50,51,52]. Eucalyptol and Camphor were the major identified constituents (50.04%, 17.75%), respectively. Additionally, monoterpenes and sesquiterpenes were the most abundant classes (Figure 3 and Figure 4 and Table 3).

### 3.4. In Vitro Anti-H. pylori Activity of the TES and the Essential oil of S. officinalis

Potential antimicrobial activity against *H. pylori* is demonstrated in both tested samples TES and oil (Table 4 and Figure 5) compared to those of the standard used medication clarithromycin (MIC 0.48 g/mL). The highest activity was revealed by the TES (MIC 3.9 g/mL). However, oil displayed only modest action (15.63 MIC g/mL). The differences in chemical composition between TES and the oil could be the cause of this variation. This was further confirmed using the in silico study (Table 5 and Figure 6).

### 3.5. In Silico Evaluation of Anti-H. pylori Activity

Screening for enzyme inhibitors involved in pathogen metabolism and biosynthesis is one of the strategies performed nowadays to combat bacteria [62,63]. *H. pylori* was found to have enzymes from the pentose phosphate pathway (PPP), Glucose-6-phosphate isomerase, glucose 6-phosphate dehydrogenase (HpG6PD), and 6-phosphogluconolactonase were investigated by [64]. PPP may serve as a means of supplying NADPH for reductive processes and biosynthesis. Therefore, we focused on finding and testing compounds to inhibit the G6PD of *H. pylori* as a tactic for the rational design of drugs against this bacterium. We investigated 30 compounds that revealed HpG6PD’s inhibition. All the compounds tested by molecular docking had the potential to bind with the active site of the enzyme. This may be explained by the studied compounds’ potential to interact with the essential amino acids forming the enzyme through H-bonding and alkyl interactions (Figure 6). Notably, most of the phenolic compounds and flavonoids identified from LC/MS have exceeded the fitting energy of co-crystallized ligand (NADP) nicotinamide-adenine-dinucleotide phosphate (−29.6914 kcal/mol) suggesting possessing good inhibiting activity against the HpG6PD. This explains the potential of TES in anti-*H. pylori* activity, which could be related to its main components. On the other hand, the essential oil’s main compounds had the same or even weaker fitting energy score compared to the co-crystallized ligand (NADP), which explains the moderate in vitro anti-*H. pylori* activity of the oil (Table 5).

### 3.6. COX-2 Inhibition Assay (Anti-Inflammatory Assay)

Given that *H. pylori* are frequently linked to a wide range of inflammations, the potential anti-inflammatory activity of both tested samples was assessed. Inflammation caused by the persistent infection can, in most cases, remain undiagnosed. Gastric inflammation, however, can develop into gastritis, gastric mucosa-associated lymphoid tissue MALT) lymphoma, peptic ulcer, and gastric cancer in some of the *H. pylori*-infected population [65]. The inhibition percentage using COX-2 assay was expressed as mean standard deviation, as seen in (Table 6 and Figure 7). The results revealed that, compared to the standard medication, celecoxib (positive control) had a IC_50_ = 0.43 ± 0.12. The two tested samples inhibited COX-2 expression with various levels (Figure 5), and TES revealed a strong anti-inflammatory activity approaching celecoxib with IC_50_ = 1.78 ± 0.27. However, the oil had a weaker activity with IC_50_ = 5.3 ± 0.62, confirming that the variation in the active metabolites resulted in a difference in the anti-inflammatory activity.

## 4. Discussion

*H. pylori* infection is linked with numerous distinct chains of diseases in Egypt and other developing countries; however, this is not commonly seen in industrialized societies. The prevalence of *H. pylori* infection among teenagers in the United States pales in comparison to infection rates of young children at 5 years of age in some regions of the developing world. While *H. pylori* are associated with gastritis, which can eventually lead to gastric carcinoma, and peptic ulcers, the infection appears to be linked with chronic diarrhea, malnutrition, and growth failure [66,67], as well as a predisposition to other enteric infections such as typhoid fever and cholera [68]. Treatment of *H. pylori* is seen as an additional challenge due to economic constraints such as the test-and-treat strategy, which lays a significant economic burden on many of these countries, as well as the prevalence of antibiotic resistance and poor patient compliance [69]. Proton pump inhibitor (PPI)-clarithromycin-amoxicillin or metronidazole treatment for two weeks is suggested as the first-line treatment for *H. pylori* infection [59,70,71]. Doctors usually prescribe anti-*H. pylori* therapy with an eradication rate of 90% at least for the treatment. However, multiple extensive clinical studies have revealed that the usual triple therapy eradication rate has generally decreased to unacceptable levels (i.e., less than 80%). Success rates in several countries are regrettably declining to reach as low as 25% [70]. The causes for this decline in efficacy over time are unknown, however, it may be connected to the rising prevalence of clarithromycin-resistant strains of *H. pylori* [70]. Thus, it was deemed necessary to search for novel therapeutic regimens that can be used as an extra constituent of antibiotic therapy or may replace current antibiotic treatments [72].

Plants have been a major source of novel pharmaceuticals and traditional treatments. They possess active metabolites that can be used to treat a variety of infectious diseases with little or no toxicity. The number of studies on medicinal plants and their potential as *H. pylori* agents has grown significantly in recent years [73,74,75]. Awareness of the use of medicinal plant metabolites as prophylaxis and therapeutics over synthetic drugs is now growing worldwide [74].

*The Salvia* genus is well known to possess significant pharmacological activity in the context of ethnopharmacological knowledge, especially in the treatment of bacterial infections [76]. The antimicrobial activity of *S. officinalis* was studied decades ago and was attributed to the presence of many active secondary metabolites [77].

In the current work, we focused on the chemical profile of the TES and essential oil, along with assessing the anti-*H. pylori* and anti-inflammatory effects. The results revealed a great variation in the chemical profile of both tested samples. The terpenoids (carnosic acid, β-sitosterol, rosmadial, 12-*O*-methyl carnosol, and carnosol); phenolic acids (rosmarinic acid); polyphenols (rosmanol and epirosmanol); and Flavonoids (dihydroxy kaempferol, Hispidulin, and cirsimaritin) were the dominant constituents in the TES.

However, monoterpenoids (eucalyptol, camphor, α-pinene, camphene, limonene, β-myrcene, α-terpineol, α-thujone, camphor, E-pinocamphone, endo borneol, myrtenol, and bornyl acetate) and sesquiterpenoids (caryophyllene, caryophyllene oxide, α-humulene, and viridiflorol) were dominant in the essential oil (Table 3). Due to the difference in the chemical profile, the two tested samples exhibited variation in biological activity. The TES displayed more potent anti-*H. pylori* activity is asymptotic to that of the standard antibiotic clarithromycin, in addition to a high percentage of inhibition of the COX-2 enzyme, indicating greater potential as an anti-inflammatory agent. This potent activity may be related to the active metabolites present in the TES.

According to the literature, the active metabolite carnosic acid and its derivatives are reported to have broad antimicrobial activity against Gram-positive and Gram-negative bacteria. It has been suggested that they exert their activity because of the lipophilic properties of these compounds allowing them to penetrate the bacterial membrane and interact with the membrane phosphorylated groups via their hydrogen bond-donor group (s) [78,79]. Moreover, carnosic acid and its derivatives may act as a modulator of the efflux pump through the dispersion of the membrane potential [78,79,80]. Despite showing moderate effect when used alone, synergisms were reported when added to antibiotics such as Erythromycin and tetracycline, improving their effectiveness by up to 8- and 16-fold against *S. Aureus* and, *E. faecium*, and *E. faecalis*, respectively [78]. On the other hand, carnosic acid demonstrates anti-inflammatory activity via stimulating the peroxisome proliferator-activated gamma receptor (PPARγ) that modulates the genetic expression of inflammatory mediators. Furthermore, they act by inhibiting the formation of pro-inflammatory leukotrienes and the secretion of human leukocyte elastase, in addition to attenuating the formation of reactive oxygen species (ROS) [81,82,83]. Rosmarinic acid is another metabolite in the TES. Based on previous studies, it is considered one of the most active recorded anti-*H. pylori* phytochemicals with antioxidant and anti-inflammatory potential. It was proven to increase the activity of the antibiotic when combined with them [84]. Flavonoids found in the TES (dihydroxy kaempferol, Hispidulin, and Cirsimaritin) may contribute to the activity of the extract. Flavonoids were reported to be more active against Gram-negative bacteria [85]. They exhibit their anti-inflammatory effect via multiple pathways, including modulation of inflammatory signaling, reduction of inflammatory molecule production, decreased attraction and activation of inflammatory cells, regulation of cellular function, and antioxidative activity [86]. Despite the limited solubility of β-sitosterol, it was documented to possess wide antibacterial activity [87] and to inhibit the generation of inflammatory cytokines and partial inhibition of NF-κB in macrophages [88]. All those active metabolites in the TES may contribute to the pharmacological activity of the extract observed in the current study.

The oil demonstrated moderate antimicrobial activity as it contains volatile monoterpenes with eucalyptol and camphor being the major compounds. Both compounds are confirmed to possess antibacterial activity by [89]. Nonetheless, the oil demonstrated weaker anti-*H. pylori* activity, most of the compounds are well-established antioxidant agents such as limonene, pinene, and borneol [90,91]. Investigations in vivo and in vitro have revealed that compounds with high antioxidant activity, not only scavenge free radicals but also exhibit antibacterial activity against *H. pylori* [92].

To determine the potential of the identified compounds against *H. pylori*, an in silico investigation using a molecular docking technique was performed. Rosmarinic acid, β-sitosterol, and carnosic acid had the highest energy score exceeding the co-crystallized ligand (NADP). This may assist in explaining the exceptional anti-*H. pylori* efficacy of TES. The oil constituents have a similar inhibitory activity to 7SEH or even less may explain the modest action of the essential oil compared to TES. However, more in vivo investigation is needed to confirm the activity.

## 5. Conclusions

The common inadequacy of conventional antibiotics to control *H. pylori* infections has generated interest in alternative curative agents. *S. officinalis* aerial parts are widely consumed traditionally for their therapeutic importance. In our study, the plant proved to be a potentially potent drug that can be used to eradicate the *H. pylori* infection or to alleviate the inflammation associated with it. The in silico study substantiates the activity of the identified constituents.

More studies are required to evaluate the *Salvia* metabolites’ effects on bacteria, as well as how they impact antimicrobial resistance when combined with conventional antibiotics. Furthermore, in vivo research is required to develop a more simple, natural, cost-effective, and advantageous antibacterial and anti-inflammatory pharmaceutical product.

## Figures and Tables

**Figure 1 metabolites-13-00136-f001:**
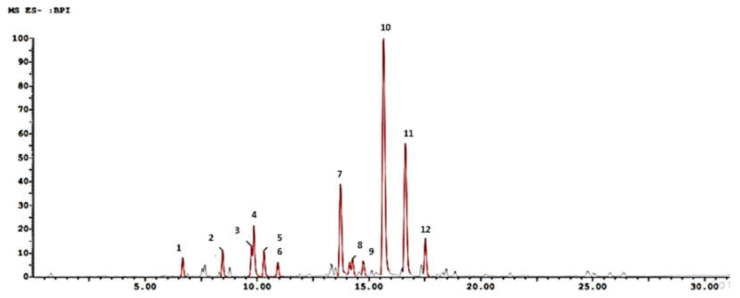
[M-H] chromatogram of the *S. officinal* ethanolic extract.

**Figure 2 metabolites-13-00136-f002:**
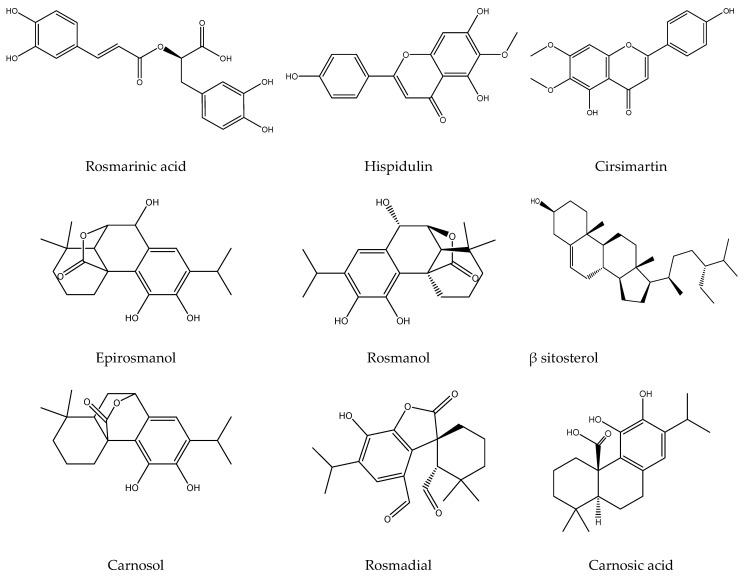
Two-dimensional structures of the identified compounds of TES by LC–MS analysis (Chem-draw ultra-version 14).

**Figure 3 metabolites-13-00136-f003:**
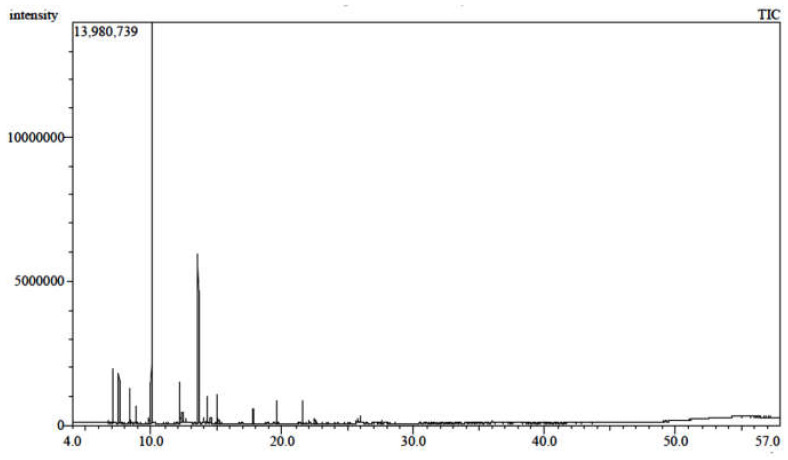
GC/MS chromatogram of the essential oil of *S. officinalis*.

**Figure 4 metabolites-13-00136-f004:**
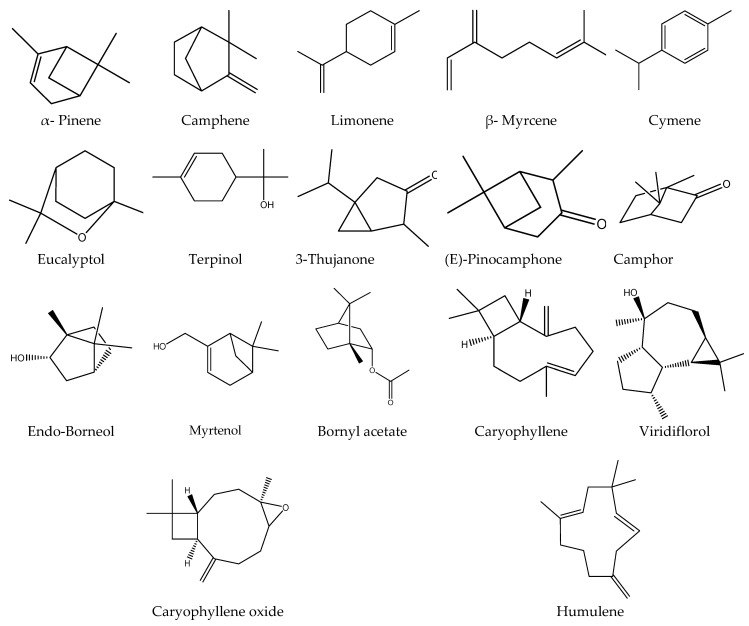
Two-dimensional structures of the compounds identified from the oil by GC/MS analysis (Drawn by Chem-draw ultra-version 14, PerkinElmer Informatics).

**Figure 5 metabolites-13-00136-f005:**
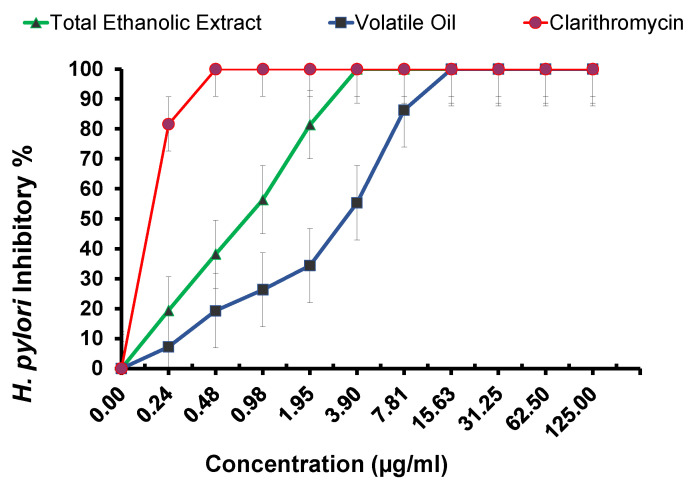
In vitro anti-*H. Pylori* activity of the total ethanol extract (TES), the essential oil of *S. officinalis*, and Clarithromycin.

**Figure 6 metabolites-13-00136-f006:**
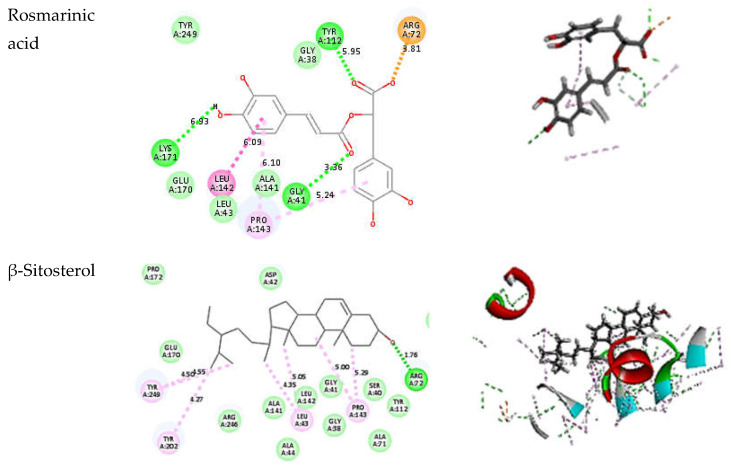
Two-dimensional and three-dimensional diagrams demonstrating the interaction of the co-crystallized ligand (NADP) and identified compounds docking pose interactions with the key amino acids in the HpG6PD-binding site. The three-letter amino acid codes plus the positions of each residue are recorded. Between the receptor and the ligand, hydrogen-bonding interactions are illustrated by a green line, whereas the purple dashed line represents alkyl interactions. Two-dimensional and three-dimensional diagrams were obtained using Discovery Studio 4.5.

**Figure 7 metabolites-13-00136-f007:**
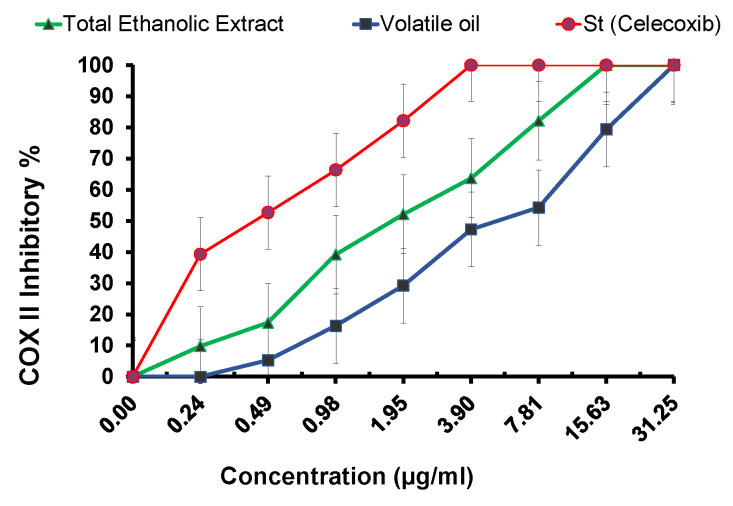
Anti-inflammatory activity cyclooxygenase COX-2 inhibitory % of TES and the essential oil compared to the standard drug celecoxib.

**Table 1 metabolites-13-00136-t001:** Preliminary phytochemical screening of *S. officinalis*.

Active Ingredients	Test	Test Procedure	Observation	Abundance
Flavonoids	NaOH	Ethanol extract + 10% NaOH + dilute HCl	The yellow solution turned colorless with the addition of dilute HCl	+++
	NH_3_	1 drop of ethanol extract is exposed to ammonia vapor	Yellow color.	
Carbohydrates	Molisch	Aqueous extract + 2 mL alcoholic α-naphthol + a few drops of concentrated H_2_SO_4_.	A Violet ring is formed	+++
Tannins	Ferric chloride.	Ethanol extract + FeCl_3_	A green solution	*+++*
Saponins	Frothing test.	Aqueous Extract + distilled water was vigorously shaken	Persistent Froth for 3 min.	*+*
Sterols	Liebermann.	Ethanol extract + 2 mL CHCl_3_ + conc. H_2_SO_4_ to form a lower layer	A reddish-brown ring at interphase	*++*
Volatile oil	Filter paper stain	Press the aerial parts between filter paper	A transient stain is formed that evaporates upon standing.	*++*

Most Abundance +++, moderate abundance ++, Traces +.

**Table 2 metabolites-13-00136-t002:** Peak assignments of the methanol extracts of *S. officinal* via LC-ESI/MS in negative ionization mode.

No	Name	Rt (min)	Molecular Formula	(M-H)	M.wt	Abundance	References
**1.**	Rosmarinic acid	6.68	C_18_H_16_O_8_	359	360	2.1%	[39]
**2.**	Hispidulin	8.45	C_16_H_12_O_6_	299	300	2.8%	[40]
**3.**	Cirsimaritin	9.76	C_17_H_14_O_6_	313	314	2.8%	[40]
**4.**	12-*O*-methyl carnosol	9.86	C_20_H_26_O_5_	345	346	6.15%	[41]
**5.**	Rosmanol	10.31	C_20_H_26_O_5_	345	346	2.92%	[42]
**6.**	β-sitosterol	10.92	C_29_H_50_O	413	414	1.72%	[43]
**7.**	Carnosol	13.73	C_17_H_14_O_7_	329	330	13.3%	[44]
**8.**	Carnosol isomer	14.45	C_17_H_14_O_7_	329	330	1.45%	[44]
**9.**	Rosmadial	14.27	C_20_H_24_O_5_	343	344	2.00%	[45]
**10.**	Carnosic acid	15.66	C_20_H_28_O_4_	331	332	37.66%	[46,47]
**11.**	Epirosmanol isomer	16.63	C_20_H_26_O_5_	345	346	20.65%	[45]
**12.**	6,8-Dihydroxykaempferol	17.53	C_15_H_10_O_8_	316	317	1.1%	[48]

**Table 3 metabolites-13-00136-t003:** The chemical profile of the identified compounds using GC/MS arranged according to the retention times.

Peak No	Rt	Compound	KIe	KIl	Area %	Reference
1	7.120	α-Pinene	934	932	3.99	[53]
2	7.540	Camphene	949	946	3.80	[53]
3	8.374	Limonene	1051	1049	2.66	[53]
4	8.844	β-Myrcene	990	988	1.29	[53]
5	9.861	Cymene	1024	1020	0.52	[53]
6	10.086	**Eucalyptol (1,8-Cineole)**	1029	1025	**50.04**	[53]
7	12.207	α-terpineol	1188	1187	3.62	[53]
8	12.358	3-Thujanone	1124	1121	0.97	[54]
9	12.700	α-Thujone	1102	1100	0.34	[53]
10	13.581	**Camphor**	1146	1143	**17.75**	[55]
11	14.056	(E)-Pinocamphone	1163	1158	0.50	[25]
12	14.252	Endo-Borneol	1169	1165	3.26	[25]
13	14.587	4-Terpinol	1177	1176	0.56	[56]
14	15.007	α-Terpineol	1188	1184	2.78	[57]
15	15.187	Myrtenol	1195	1193	0.44	[58]
16	17.801	Bornyl acetate	1287	1286	1.30	[25]
18	21.558	Caryophyllene	1419	1422	2.16	[59]
19	22.464	Humulene	1457	1452	0.46	[60]
20	25.790	Caryophyllene oxide	1583	1582	0.41	[25]
21	26.010	Viridiflorol	1592	1590	0.84	[61]
Total identified components					95.89%	
Monoterpene Hydrocarbon					12.21%	
Oxygenated monoterpene					81.56%	
Oxygenated sesquiterpene					1.66%	
Sesquiterpene Hydrocarbon					0.46%	

Kovacs indices were determined experimentally (KIE) on a HP-5MS column relative to C_8_–C_28_ n-alkanes. KIL, literature published Kovacs retention indices. The main compounds are in bold.

**Table 4 metabolites-13-00136-t004:** In vitro anti-*H. Pylori* activity of the total extract (TES), the essential oil, and Clarithromycin.

Sample Concentration (µg/ mL)	Total Ethanolic Extract		Volatile Oil		Clarithromycin	
	Mean of *H. pylori* Inhibitory %	S.D.	Mean of *H. pylori* Inhibitory %	S.D.	Mean of *H. pylori* Inhibitory %	S.D.
125	100	-	100	-	100	-
62.5	100	-	100	-	100	-
31.25	100	-	100	-	100	-
15.63	100	-	100	-	100	-
7.81	100	-	86.32	1.5	100	-
3.9	100	-	55.34	2.4	100	-
1.95	81.42	0.85	34.38	1.3	100	-
0.98	56.37	1.1	26.34	0.69	100	-
0.48	38.23	0.63	19.3	0.95	100	-
0.24	19.38	1.4	7.2	0.83	81.64	0.58
0	0	-	0	-	0	-
MIC	3.9		15.63		0.48	

All determinations were carried out in triplicate, and values are expressed as the mean ± SD.

**Table 5 metabolites-13-00136-t005:** Free binding energies (∆G) of the compounds identified within the HpG6PD active site calculated in kcal/mol using Discovery Studio 4.5 adopting both rule-based ionization techniques.

TES Compounds		Oil Compounds	
Compound	Binding Energy ∆G (Kcal/mol) Rule-Base	Compound	Binding Energy ∆G (Kcal/mol) Rule-Base
Rosmarinic acid	−46.6769	co-crystallized ligand (NADP)	−29.6914
β-Sitosterol	−44.8608	Borneyl acetate	−29.2608
Carnosic acid	−40.7992	α-Terpineol	−25.2219
Epirosmanol	−40.4131	Caryophyllene	−24.4108
12-*O*-methyl carnosic acid	−39.6734	(E)-Pinocamphone	−23.0788
Carnosol	−38.5699	4-Terpineol	−22.6625
Rosmanol	−34.6064	Myrtenol	−22.6432
Dihydroxy kamepferol	−34.5906	Thujanone	−22.5693
Rosmadial	−34.5739	Endoborneol	−22.497
Cirsimaritin	−33.5894	Camphor	−21.7282
Hispidulin	−33.1293	Cymene	−20.7157
co-crystallized ligand (NADP)	−29.6914	Myrcene	−20.2976
		Eucalyptol	−19.1464
		Limonene	−19.1059
		α-Pinene	−18.6418

**Table 6 metabolites-13-00136-t006:** COX-2 inhibition assay of TES and the essential oil compared to the standard drug celecoxib.

Sample Conc. (µg/mL)	Total Ethanolic Extract		Volatile Oil		St (Celecoxib)	
	Mean of COX-2 Inhibitory %	S.D.	Mean of COX-2 Inhibitory %	S.D.	Mean of COX-2 Inhibitory %	S.D.
31.25	100.00	-	100.00	-	100	-
15.63	100.00	-	79.35	0.85	100	-
7.81	82.17	0.71	54.31	0.74	100	-
3.9	63.74	0.89	47.32	0.34	100	-
1.95	52.19	1.3	29.31	0.82	82.15	0.63
0.98	39.21	0.96	16.35	0.14	66.34	1.2
0.49	17.35	0.74	5.31	0.52	52.72	0.58
0.24	9.85	0.82	0.00	-	39.35	1.2
0.00	0.00	-	0.00	-	0.00	-
IC_50_	1.79 ± 0.27		5.3 ± 0.62		0.43 ± 0.12	

## Data Availability

Not applicable.

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
