# Peer review of "Anti-*Heliobacter pylori* and Anti-Inflammatory Potential of *Salvia officinalis* Metabolites: In Vitro and In Silico Studies"

_metabolites, 2023, doi:10.3390/metabo13010136_

Round 1
Reviewer 1 Report
Phytochemical screening, LC-MS, GC-MS, Anti-Heliobacter- Pylori & anti-inflammatory potential of Salvia officinalis (In-Vitro and In-silico studies). The study provides phytochemical analysis and biological potential of Salvia officinalis, which will be helpful for the research on metabolites. However, there are some deficiencies which must be addressed.
The title should be revise and do not use symbols in the title.
Specific quantitative results must be provided in the abstract.
Add full form of abbreviations at first use.
Add conclusion and future perspective in abstract.
Add reference here “remedies for at least 60,000 years”
In line 38 add relevant reference “https://doi.org/10.1016/j.chnaes.2021.08.002”
Add more details of the economic and medicinal importance of salvia, specifically antimicrobial significance.
Section 2.7 line 156 should be cited with relevant study. “https://doi.org/10.3390/coatings12101505”
Figure 5 how 2 D and 3 D diagrams were obtained.
Grammatical mistakes and typos must be revised in the whole MS.
Most important the authors should be consistent using abbreviations in the whole text.
Conclusion looks like a background of the study. Conclusion must be findings based and proposing gaps and future recommendations.
Author Response
Responses to Reviewer 1 Comments
Phytochemical screening, LC-MS, GC-MS, Anti-Heliobacter- Pylori & anti-inflammatory potential of Salvia officinalis (In-Vitro and In-silico studies). The study provides phytochemical analysis and biological potential of Salvia officinalis, which will be helpful for the research on metabolites. However, there are some deficiencies which must be addressed.
We would like to express our deep thanks for the reviewers` opinions and we hope that we can fit all the required corrections, blue highlights represent the responses.
Point 1: The title should be revise and do not use symbols in the title.
Response 1:
Thanks for the reviewers` opinions. The title was checked and modified as requested.
Point 2: Specific quantitative results must be provided in the abstract.
Response 2:
We thank the reviewer for such a valuable remark. The quantitative results are added.
Point 3: Add full form of abbreviations at first use.
Response 3:
We apologize for these unintentional errors; the entire manuscript is revised in this regard.
Point 4: Add conclusion and future perspective in abstract.
Response 4:
We appreciate the reviewer's thoughtful comment; please check the abstract modifications done as recommended.
Point 5: Add reference here “remedies for at least 60,000 years”
In line 38 add relevant reference https://doi.org/10.1016/j.chnaes.2021.08.002
Response 5:
Done, the reference is added as recommended.
Point 6: Add more details of the economic and medicinal importance of salvia, specifically antimicrobial significance.
Response 6
We would like to express our gratitude for the reviewer's comment. More detailed statements are added as requested.
Point 7: Section 2.7 line 156 should be cited with relevant study. “https://doi.org/10.3390/coatings12101505”
Response 7:
Done, the reference is added as suggested.
Point 8: Figure 5 how 2 D and 3 D diagrams were obtained.
Response 8:
We appreciated this insightful feedback. The program name is already mentioned in the Experimental Design section part 2.8. The program name is added under the figure as requested.
Point 9: Grammatical mistakes and typos must be revised in the whole MS.
Response 9:
We express regret for this typographical error. Revised and corrected.
Point 10: Most important the authors should be consistent using abbreviations in the whole text.
Response 10:
Thanks for the priceless advice, the abbreviation was rechecked.
Point 11: Conclusion looks like a background of the study. Conclusion must be findings based and proposing gaps and future recommendations.
Response 11:
We appreciated this insightful feedback. The conclusion was edited and future recommendations were added.
Reviewer 2 Report
The manuscript entitled "Phytochemical screening, LC-MS, GC-MS, Anti-Heliobacter-Pylori &;anti-inflammatory potential of Salvia officinalis (In-Vitro and In-silico studies)" evaluated the chemical profile of the total ethanolic extract and the essential oil of Salvia officinalis to explore the therapeutic potential on the anti-H. pylori and the anti-inflammatory activities. It has been documented as a potential potent drug against H. pylori to eradicate the bacterium or alleviate the inflammation associated with it. It is a routine work but with some promises. However some of my observations are given as follows for the overall improvement of the essence of the text:
1.Please check your title.
One suggested title may be:
Anti-Heliobacter-Pylori &anti-inflammatory potential of Salvia officinalis :In-Vitro and In-silico studies.
2.In third line of 'Abstract'(16th line), ".....................in this regard" may be used in place of "In this study".
3. Suitable Reference /(s) needed to supplement the statement/(s) appeared in line #s 47, 56, 292, 304, etc.
4. In 2.1, line # 96-98 may be deleted and can be incorporated in the 'Acknowledgement' section.
5. Some minor mistakes may be corrected like ........replacement of "gm" with "g", avoiding usage of complex terminologies and repetition of letters/words as in line #166-167, 174, etc.
6.In the entire "Discussion", there are a few short sentences as new paragraph(s). It needs proper attention and suggested to rewrite this important section so that it becomes scientifically sound as well as logically competent for the readers. Citation of results in this section should be minimized to avoid repetition.
7. In line # 384-386, the statement may be rewritten for clarity.
8. Altogether, the conclusion must be comprehensive and should reveal future scope as well.
The above corrections must be addressed immediately.
Author Response
Response to Reviewer 2 Comments
The manuscript entitled "Phytochemical screening, LC-MS, GC-MS, Anti-Heliobacter-Pylori &; anti-inflammatory potential of Salvia officinalis (In-Vitro and In-silico studies)" evaluated the chemical profile of the total ethanolic extract and the essential oil of Salvia officinalis to explore the therapeutic potential on the anti-H. pylori and the anti-inflammatory activities. It has been documented as a potentially potent drug against H. pylori to eradicate the bacterium or alleviate the inflammation associated with it. It is routine work but with some promises. However, some of my observations are given as follows for the overall improvement of the essence of the text.
Response:
We would like to express our deep thanks for the reviewers` opinions and we hope that we can fit all the required corrections; all required modifications were green highlighted.
Point 1: Please check your title.
One suggested title may be:
Anti-Heliobacter-Pylori &anti-inflammatory potential of Salvia officinalis: In-Vitro and In-silico studies.
Response 1:
We would like to express our gratitude for the reviewer's input. The title was changed and the suggested one “Anti-Heliobacter. pylori & anti-inflammatory potential of Salvia officinalis metabolites; in-vitro and in-silico studies” was selected instead.
Point 2: In the third line of 'Abstract'(16th line), ".....................in this regard" may be used in place of "In this study".
Response 2:
We thank the reviewer for such a valuable remark, the sentence is added as suggested.
Point 3: Suitable Reference /(s) needed to supplement the statement/(s) appeared in lines #s 47, 56, 292, 304, etc.
Response 3:
We thank the reviewer for such a great comment, and the references are added.
Point 4: In 2.1, line # 96-98 may be deleted and can be incorporated in the 'Acknowledgement' section.
Response 4:
We appreciate the reviewer's thoughtful comment; please check the modification done as recommended.
Point 5: Some minor mistakes may be corrected like ........replacement of "gm" with "g", avoiding usage of complex terminologies and repetition of letters/words as in line #166-167, 174, etc.
Response 5:
We apologize for these typographic errors; the entire manuscript is revised in this regard.
Point 6: In the entire "Discussion", there are a few short sentences as new paragraph(s). It needs proper attention and suggested to rewrite this important section so that it becomes scientifically sound as well as logically competent for the readers. Citation of results in this section should be minimized to avoid repetition.
Response 6:
We would like to express our gratitude for the reviewer's input. The discussion is adjusted as recommended.
Point 7: In line # 384-386, the statement may be rewritten for clarity.
Response 7:
We would like to express our gratitude for the reviewer's input. The statement is rewritten for clarity as requested.
Point 8: Altogether, the conclusion must be comprehensive and should reveal future scope as well.
Response 8:
We appreciated this insightful feedback. The conclusion is edited as requested with future scopes added.
